# The Nature of the Spark Is a Pivotal Element in the Design of a Miller–Urey Experiment

**DOI:** 10.3390/life13112201

**Published:** 2023-11-12

**Authors:** Sina Ravanbodshirazi, Timothée Boutfol, Neda Safaridehkohneh, Marc Finkler, Mina Mohammadi-Kambs, Albrecht Ott

**Affiliations:** Biological Experimental Physics, Center for Biophysics, Faculity of Natural Sciences and Technology, Saarland University, Campus B2 1, 66123 Saarbrücken, Germany; sina.ravanbodshirazi@physik.uni-saarland.de (S.R.);

**Keywords:** origin of life, Miller–Urey, GC/MS, spark

## Abstract

Miller and Urey applied electric sparks to a reducive mixture of CH_4_, NH_3_, and water to obtain a complex organic mixture including biomolecules. In this study, we examined the impact of temperature, initial pressure, ammonia concentration, and the spark generator on the chemical profile of a Miller–Urey-type prebiotic broth. We analyzed the broth composition using Gas Chromatography combined with Mass Spectroscopy (GC/MS). The results point towards strong compositional changes with the nature of the spark. Ammonia exhibited catalytic properties even with non-nitrogen-containing compounds. A more elevated temperature led to a higher variety of substances. We conclude that to reproduce such a broth as well as possible, all the studied parameters need to be tightly controlled, the most difficult and important being spark generation.

## 1. Introduction

In 1953, Stanley L. Miller and Harold C. Urey exposed water, methane, ammonia, and hydrogen to an electric spark, demonstrating the non-biological synthesis of biologically relevant compounds [1]. Their work paved the way for research into the development of life on early Earth. Cleaves et al. demonstrated the significance of pH for prebiotic chemistry [2]. Both climate and pH are controlled by the carbon cycle [3,4]. The pH of the Earth’s oceans evolved from 6.3–7.2 at 4.0 Ga, to 6.5–7.7 at 2.5 Ga and 8.2 at the present time. This is primarily driven by the secular decline of p CO_2_ (the partial pressure of carbon dioxide). The decline in p CO_2_ is a consequence of increasing solar luminosity. However, the evolution of the pH of the oceans is also moderated by carbonate alkalinity delivered from continental and seafloor weathering [5]. Russell et al. proposed that the origin of life occurred in alkaline hydrothermal vents resembling those found in the Lost City, a hydrothermal field in the Atlantic ocean where a proton gradient produces a natural proton-motive force at the boundary between the alkaline containing SH^−^ ions hydrothermal spring and the slightly acidic ocean (pH 5–6) containing iron in the water [6,7]. Regarding the emergence of macromolecules, dry conditions at high temperature and high pressure promote abiotic polymerization [8,9]. The catalytic role of transitional metals like Ni/Fe and iron sulfide (FeS) could facilitate the reduction of N_2_ to NH_4_+ in an aqueous solution [10,11]. All cellular substances may well have evolved at the same time through common chemistry [12].

The latter finding sparked new interest in the generation of prebiotic broths as produced via the Miller–Urey experiment. Different gas mixtures increase the diversity of the observed chemical compounds [2,13,14,15,16]. Different energy sources such as UV [17,18], X-ray [19,20], laser photolysis [21], and high-energy proton irradiation can replace the spark [22]. Experiments suggested that complex molecules including amino acid precursors were formed from simple molecules like HCN in the gas phase [22]. In a recent study, Mohammadi et al. proposed that the accumulation of formic acid as an intermediate of Fischer–Tropsch synthesis may well have occurred on early Earth through multiple, independent pathways [23,24]. They note that ammonium salts of formic acid have been suggested to serve as the precursor of formamide. Ferus et al. revisited the Miller–Urey experiments by using electric discharge and laser-driven plasma simulations in a reducing atmosphere to investigate nucleobase formation from formamide [25]. The synthesis of organic compounds through the action of electric discharges on neutral gas mixture conditions has been shown to be much less efficient compared to the reducing gas mixture of the original experimental setup. However, significant amounts of amino acids were produced even from neutral gas mixtures [2]. The presence of borosilicate glass as a catalyst in the reaction vessel significantly increased the molecular diversity and yield of organic matter [26]. The production of molecular species that are typically not preferred under reactor conditions such as polyethyleneglycol suggests the existence of unidentified organocatalysts [27].

Theoretical computational approaches have been developed as well. A reactive machine learning system combined with a nanoreactor molecular dynamics’ sampler was used to simulate the behavior of C, H, N, and O elements in a wide range of real-world reactive systems. It closely matched experimental structures in carbon solid-phase nucleation and graphene ring formation studies and produces reliable predictions in cases where experimental data are not available [28]. A different ab initio nanoreactor simulation suggested new pathways for glycine synthesis from primitive compounds proposed to exist on early Earth. However, such a nanoreactor does not aim to replicate the physicochemical conditions of any specific environment [29]. A theoretical study at the quantum level based on ab initio molecular dynamics showed that, rather than the activation energy of electric discharge, it was the strong electric field that favored the formation of small intermediate molecules such as formic acid and formamide in Miller–Urey-type experiments [30].

Vincent et al. pointed out that although the concept of “prebiotic broth” has received considerable attention, there has been a notable absence of well-designed, real chemical mixtures. They addressed this gap by exploring principles and guidelines for selecting chemical mixtures, considering both assembled and synthesized options [31]. In our previous findings, we highlighted the reactor design as an important factor affecting the complex mixture of a Miller–Urey-type experiment [32]. To maintain a high level of experimental control and facilitate a comprehensive analysis of the significance of each parameter, we streamlined the design of the traditional circulating reactor to a simplified one-pot reactor. This reduced the complexity, ensuring a high level of experimental control for data comparison.

Here, we address the reactor’s physical parameters (i.e., the nature of the spark, temperature, ammonia concentration, and pressure) and their effect on composition. Note that these parameters are linked, i.e., pressure and temperature change the nature of the spark. Concentrations intervene in the pressure and the ionization of the gas mixture during sparking. To our knowledge, the above set of parameters has not been investigated before, in particular, the influence of the spark generator remains uncharacterized. We conducted GC-MS analysis to differentiate and identify the broth components’ fragmented ions to gain a better understanding of the relationship between the variation of the mentioned parameters and the resulting chemical profile.

## 2. Materials and Methods

### 2.1. Experimental Setup

We conducted four sets of experiments, focusing on temperature (samples 1, 2, and 3), pressure (samples 2 and 6), ammonia concentration (samples 4, 2, and 5) and the spark generator (samples 2, 7, and 8) (see Table 1. Experimental conditions). The experiments were performed in a 5 L flask as a reactor equipped with an overpressure valve (Normag, Hofheim-Germany) responding to a pressure of 1.3 bar (Normag, Hofheim-GermanyORMAG-Germany) as shown in Figure 1. First, 300 mL HPLC-grade water (Fisher Chemical, Loughborough-UK) were added to the flask. The 5 L flask was evacuated 3 times using a vacuum pump and subsequently filled with methane (N25 Air Liquide, Düsseldorf-Germany) to effectively remove any residual air from the flask and degas the water. Subsequently, the desired amount of ammonium hydroxide 35% *w*/*w* solution (Fisher Chemical, Loughborough-UK) was injected into the flask through a silicon septum (DWK Life Sciences, Holzminden-Germany) using a glass syringe (Fortuna Optima Luer Lock, Fisher Scientific, Schwerte-Germany) with a needle. Then, the 5 L flask was partly immersed in a bath of heated silicon oil, equipped with a magnetic stirrer. The reactor was allowed to equilibrate overnight prior to sparking. All experiments were conducted for 5 days. The electric discharge occurred in the gaseous phase between two semi-sharp electrodes. We utilized tungsten electrodes in our experiment. They were 30 cm in length, 6 mm in diameter, exhibited a length of their cone of 15 mm and a tip with 0.5 mm radius. At the opposite end of the electrodes, there was a 20 mm deep pit with a 2.5 mm internal diameter for the connection of high-voltage and earth cables through a ferrule embedded within the hole. The analysis using Energy Dispersive X-ray Spectroscopy (EDX) revealed the tungsten composition to be highly pure, with only trace amounts of impurities, notably Iron, Cobalt, Nickel, and Copper. Two flyback-based (FB-1 and FB-2) and a capacitor-based (C-1) high-voltage spark generators were used (Appendix A for details).

We compared three different ammonia concentrations in the Miller soup. The ammonia partial pressure in sample 2 displayed a twofold increase compared to sample 4 (low ammonia). The high-ammonia solution (sample 5) underwent an approximately fourfold surge relative to the low-ammonia solution (sample 4) (Appendix A). Accordingly, the ammonia-to-methane ratio in the gas phase changed; considering that the methane pressure is kept constant (except for sample 6), ammonia is responsible for the increased overall pressure.

### 2.2. Sample Preparation and Derivatization

We used the method of Bligh and Dyer [33] to extract the organic phase of our experiments. Methanol and chloroform were added to the broth to attain a final volume ratio of 2:2:1 for the chloroform/methanol/prebiotic broth. Shaking vigorously on a shaker in a separatory funnel at 150 rpm for 5 min, enabled separation so that the upper layer was no longer cloudy. The lower layer (organic phase) was drained via a stopcock embedded in the separatory funnel. The organic phase was dried at room temperature under a nitrogen purge. After drying, the samples were stored under a nitrogen atmosphere at −80 °C until analysis. Prior to GC-MS analysis, 1 mg of the sample was derivatized by adding 200 µL of the derivatizer (BSTFA + TMCS, 99:1, sylon BFT) obtained from Supelco (Bellefonte, PA, USA), followed by incubation at +70 °C for 2 h. During derivatization, the labile hydrogen is replaced by trimethyl silyl, and the molecular polarity reduces to facilitate chromatographic separation [34].

### 2.3. GC-MS Method

The samples were separated using a GC (Agilent 8890 GC, Wilmington-USA) System equipped with a 30 m capillary column (0.250 mm i.d., 0.25 µm film) HP-5MS UI fused silica capillary column (Agilent Technologies, Folsom-USA) coupled to a mass detector (5977B GC/MSD, Wilmington-USA). Then, 1µL of the derivatized sample was injected in splitless mode. The column temperature was initially held at +70 °C for 8 min, then increased to +280 °C at a rate of 3.5 °C/min with a final hold time of 9 min. Helium was the carrier gas with a constant flow rate of 0.9 mL/min. The injector temperature was maintained at +280 °C and mass spectra were scanned from 50–550 *m*/*z* at a scan rate of 0.9 scans/s; EI operated under 70 eV. For analysis, we used AMDIS software (Version 2.72) integrated with the mass-spectrometry library from the National Institute of Standards and Technology, NIST (Version 2.3, Gaithersburg, MD, USA). To exploit the relative quantification of primary formed fragments, we used the Bruker DataAnalysis software (Version 5.0, Bruker, Bremen-Germany). This software lists the intensities of all detected *m*/*z* during the time course of measurement.

## 3. Results

### 3.1. Detected Compounds

The GC-MS analysis revealed a large variety of organic compounds including alkanes (C_12_-C_44_), fatty acids, alcohols, amines, aromatics, and heterocycles among others (Table 2 and Appendix A). The detected compounds have a wide range of degrees of aromaticity and chemical variability as shown in Table 2 and Appendix A. We identified several compounds that were common in all samples, regardless of the applied conditions, e.g., alkanes, fatty acids and carbamates. However, the samples showed major differences in composition with respect to the experimental conditions.

Table 2 shows that Guanine and ethanimidic acid consistently formed in the prebiotic broths except in sample 1 (80 °C). Cyanophenol was not detected in samples 1 (80 °C), 2 (100 °C) and 8 (CA-1 spark generator). Oxalic acid was only observed in samples prepared at 120 °C (sample 3), or at high-ammonia concentrations (sample 5) or with spark generator FB-2 (sample 7). We also detected components such as symmetrical arrangements of ketone groups, di-ketones, and bisphenols, which can be considered radical traps for oxidation. 1,4-benzoquinone was identified in all samples, except for sample 4 (lower concentration of ammonia).

Biphenyldiol was observed in the sample with higher ammonia/methane ratios (samples 5 and 6) as well as with FB-2 (sample 7). Short polyethylene glycol (PEG) strands were detected in samples 1, 3, and 8, while the possible corresponding monomer, ethylene glycol, was detected in samples 1, 2, 3, 5, 6, and 7. A distinct set of polycyclic aromatic hydrocarbons (PAHs) was observed in samples 5, 7, and 8.

Shortly after the start of the electric discharge, an oil layer formed on top of the aqueous phase, while a thin deposit of black material started to form around the tip of the electrode. This deposit exhibited a porous structural configuration and continuously underwent detachment from the electrodes, descending onto the oil layer. The thickness was highly dependent on the temperature and spark generator. We observed a significant rise in the production of black material deposit at low temperature (sample 1); conversely, at high temperature (sample 3), a negligible amount of black material formed. The utilization of the FB-2 spark generator (sample 7) resulted in a significant increase in black matter (compared to FB-1 and CA-1), whereas sample 8, delivered via the capacitor-based spark generator (CA-1), produced a minimal amount of black dust. All other samples exhibited an almost comparable amount of black matter regardless of the altered parameters.

### 3.2. Fragments Intensity Analysis

The ten major fragments of each sample are shown in Table 3. In Table 4, we present the chemical structure as identified via the NIST database and in the literature. Detected fragments consist of saturated hydrocarbons, fatty acids, esters, fatty alcohols, ketones, ethers, aldehydes, sterols, ethylene glycol, and phenols, among others. The fragments 43, 57, 71, and 85 *m*/*z* were attributed to hydrocarbon. The fragments 341, 313, 147, 145, 132, 129, 117, and 45 *m*/*z* were attributed to fatty acids and fatty alcohols; 180, 166, and 165 *m*/*z* were attributed to phenol components. Some components were not among common molecule fragmentation products found in the literature, and we assigned these molecular fragments according to the GC-MS database, e.g., 330, 263, 222, and 221 *m*/*z*. Moreover, due to the derivatization, we found some ions that originated from the BSTFA removal of the labile hydrogens. Quantitatively, fatty acid fragments were the most abundant ones, except for sample 6, where the phenol motif dominated. In Table 3, we identified a strong similarity between samples 2 and 3, differing in temperature only (100 °C and 120 °C). The fragmentation pattern of sample 1 (produced at 80 °C) was similar but did not exactly follow the same path.

Fragments 117, 75 and 73 *m*/*z* (fatty acids, fatty alcohols, Ethers, and Aldehydes) were present in samples 2, 4 and 5. In low- and mid-ammonia concentrations, three more fragments were common (132, 57, and 43 *m*/*z*). These fragments were not among the top ten of the high-ammonia sample (sample 5).

Reducing the pressure to 0.7 bars is achieved by decreasing the level of methane. Comparing sample 2 (at atmospheric pressure) and sample 6 (at 0.7 bars), we found that they exhibited the same three common fragments, 117, 73, and, 75 *m*/*z*, which correspond to fatty acids, fatty alcohols, ethers, and aldehydes. Notably, fragment 73 *m*/*z*, representing ethers and aldehydes, held the second-highest rank. In the low-pressure condition (sample 6), the chromatogram was dominated by fragment 165 *m*/*z*, associated with phenols.

Three distinct spark generators, FB-1, FB-2, and CA-1, were employed for the synthesis of samples 2, 7, and 8. For both flyback-based spark generators (FB-1 and FB-2), ions with *m*/*z* values of 145, 132, 129, 117, 75, 73, 57 and 43 (fatty acids, fatty alcohols, ethers, aldehydes, and hydrocarbons) ranked among the top ten most abundant ions. Conversely, sample 8, produced using spark generator CA-1, displayed the top five most abundant ions 165, 147, 180, 45, and 166 *m*/*z* (phenols, fatty acids, fatty alcohols, PEG, ethers, dicarboxylic acids, bisphenols, and ketones). However, ions with 165, 147, 180 and 166 (*m*/*z*) appeared in samples 5 and 6 among the top 10, suggesting that the CA-1 spark generator would (at least partly) compensate for lower ammonia/methane ratios.

### 3.3. Limitations of Our Analytical Method

One significant limitation pertains to the vaporization of the analyte, which we improved through derivatization techniques. Another challenge concerns the effective separation of compounds within the GC-MS system.

We used a HP-5MS UI fused silica capillary column (Agilent Technologies). It is non-polar and will not provide optimal separation for polar compounds. However, developing a non-targeted comprehensive method for all compounds is impossible. Moreover, we expect several factors to contribute to a weak detection of certain compounds in our study, such as for instance formic acid or amino acids (see Table 2 and Appendix A), which were detected elsewhere by others [60,61,62].

Method and Run Time: The five-day duration of our experiment may have impacted the composition by altering the ratios of components. Different run times can lead to variations in product yields and product composition.Extraction Procedure: the use of chloroform, a non-polar solvent, for extraction may exclude or reduce the recovery of highly polar compounds.Chemical Reactions: during the drying or extraction process, unintended reactions could have occurred, potentially altering the composition in both the water-methanol and chloroform phases [33,63].GC-MS Method: The wide range of compounds generated in our experiment posed a challenge for analysis. To ensure clarity and achieve robust results, we focused on non-polar compounds with *m*/*z* values between 50 and 550. This allowed us to report compounds with acceptable signal-to-noise ratios while ignoring peaks that did not meet our predefined standards. This occurred in situations where substances could not be separated by the column because their migration properties were too similar.

## 4. Discussion

Studying temperature, we observed that black material formation is maximized by lower temperature (80 °C) as well as by spark generator FB-2 compared to FB-1 and CA-1. The variety of the detected compounds was increased in sample 3 (high temperature) and in sample 7 (FB-2 spark generator) (Table 2 and Appendix A). FB-2 exhibited a capacity to generate a broader range of compounds (300 components vs. around 150–200 assigned components for all other samples (see Appendix A). However, the conditions for the highest product weight yield (15.6 mg) include spark generator FB-1, setting the temperature at 100 °C, maintaining an ammonia concentration of 5.8 g/L, and operating under standard atmospheric pressure.

Different types of spark generators generated different fragments. Sample 8, synthesized via spark generator CA-1, presents fragments 165, 147, 180, 45, and 166 *m*/*z* (phenols, fatty acids, fatty alcohols, PEG, ethers, dicarboxylic acids, bisphenols, and Ketones) as the top five abundant ions, which are not within the top ten abundant ions of the flyback-based spark generators FB-1 and FB-2 (samples 2 and 7, respectively). However, this could be compensated by increasing the amount of ammonia (sample 5). Analyzing the emission spectra of ammonia, the FB-2 spark generator leads to notably higher relative emission energy of the UV emission lines (Appendix A). We conclude that this spark generator excites the high energy levels of ammonia particularly well. This may lead well lead to the broader variety of compounds compared to the other spark generators.

Examining the effects of pressure, the analysis of the ion fragments shows that in the low-pressure condition (sample 6), there is an evident preeminence of the fragment with a mass-to-charge ratio of 165 (corresponding to phenolic compounds). This is also observed at high-ammonia conditions (sample 5), as well as in sample 8, generated via the capacitor spark generator. The reduction in the initial pressure to 0.7 bar signifies a reduction in the availability of methane, consequently amplifying the exposure of ammonia and water fractions to plasma. Higher ammonia-to-methane ratios enhance or catalyze the formation of phenolic compounds. FB-2 spark generator (sample 7) and higher ammonia concentration (sample 5) presented higher potential to produce a higher variety of the compounds. We conclude that the spark has the largest impact on the broth, besides the methane/ammonia ratio.

Considering the substances detected in GC-MS, high-ammonia concentrations (sample 5) resulted in the formation of a unique range of PAHs and unsaturated compounds compared to mid-ammonia concentrations (sample 2) (Table 2 and Appendix A). This again suggests a possible catalytic role of ammonia in the generation of unsaturated compounds. Moreover, in these conditions, the structural variety of non-nitrogen-containing compounds was enhanced, e.g., phenol derivatives, oxalic acid, PAHs and more (Table 2 and Appendix A). This points, again, towards a catalytic role of ammonia.

Carbamate is a product formed through the reaction between carbon dioxide and ammonia without needing a catalyst. Carbamate is assumed to have a crucial role in facilitating subsequent reactions in the aqueous phase, particularly due to the presence of ammonia [64]. It has been proposed that polycyclic aromatic hydrocarbons (PAHs) were delivered to Earth through meteorites [65], but our findings indicate that PAHs could have been generated autogenously in some of the experimental conditions, such as samples 5, 7, and 8. PAHs have the potential to fulfil a variety of functions in prebiotic chemistry; for instance, amphiphilic PAHs may well increase the resistance of vesicles to divalent cations [66]. PAHs may have functioned as pigments to drive photochemical reactions. They are also believed to potentially catalyze biomolecule polymerization and participate in protocell formation [67,68].

The process of auto-oxidation occurring under elevated pH conditions facilitates the production of quinones from polyphenols, widely known as potent antioxidants [69,70,71]. Subsequently, the generated quinones can undergo reduction to semiquinones, accompanied by the release of superoxides [69,70]. This unique reactivity is due to the highly alkaline nature of the reaction medium (pH 12 ± 0.5), enabling diketones to exhibit oxidizing properties by attracting dipolar and positively charged molecules. Examples of such compounds found in the prebiotic broth include 1,4-benzoquinone and biphenyldiol.

The formation mechanism of PEG short strands in the prebiotic broth is not clear but the alcohol condensation or oxirane polymerization in organic solvent with the presence of a metal catalyst, e.g., tungsten particles (detached from the electrodes), could be an option [27,72]. Previous studies have shown that saturated hydrocarbons may have been key to specific reactions necessary for the emergence of life, in particular, for the synthesis of membrane forming molecules such as single chain amphiphiles (SCAs) through Fischer–Tropsch-type (FTT) reactions [73,74,75,76]. A large amount of components with benzene motifs, branched and linear alkanes (C12-C44), along with fatty acids, amines, and solid particles from spark plasma (cf. black material), formed an interface between gas and aqueous phases [32]. Many molecules of the oil phase, such as fatty alcohols and fatty acids (Table 2), contained oxygen. Such compounds combine a hydrophobic and a hydrophilic part. They act as tensioactives. Tensioactive molecules are capable of self-assembling into membrane-bound vesicles. They accumulate at the oil–water interface, lowering the surface tension. This may be understood as a first step towards the emergence of protocells [77].

The thickness and color of the oil layer that formed between the water and gas phase varied depending on experimental parameters. An increase in the amount of hydrophobic components leads to a greater volume of the hydrophobic phase, forming a more accessible environment for reactions that predominantly occur in non-polar and aprotic solvents.

Micro- and nanostructures of metallic and oxidized tungsten originating from the electrode, incorporated with black material emerging as particles in spark plasma, spreading along the interfaces as well as in the aqueous phase are likely to enhance the catalytic properties of the broth [78,79].

The question of whether the Miller–Urey experiment can be linked to the conditions on early Earth in simple ways has been ongoing since the original work. Evidently, sulfur and phosphate, two of the most important elements in today’s organisms, are lacking in our setup, and they may have had a strong role [80]. However, these substances could be added in future work, albeit complicating the analysis. In principle, the gas mixture could also be changed to more oxidizing mixtures that may be closer to the atmosphere of early Earth. However, such experiments were performed before to conclude that a highly reductive atmosphere was a requirement to produce a highly complex broth [15]. In this context, Bada et al. showed that the yield of amino acids is greatly increased when oxidation inhibitors such as ferrous iron are added prior to hydrolysis to counteract a more oxidizing gas mixture [2]. The conclusion that locally reduced conditions were sufficient for the generation of biomolecules challenged the belief that the Earth’s atmosphere was just too oxidizing for a prebiotic broth to form spontaneously. Moreover, the oxidizing, neutral, or even slightly reductive nature of the Earth’s atmosphere, as well as of its surface, 4–4.5 × 10^9^ years ago in a prebiotic world, as well as beyond that date, remains under discussion [81,82,83].

We concur with the findings of Mohammadi et al. [22] and the suggested role of volcanic, electric discharges. Nevertheless, our sparks cannot directly be compared to lightning, however, they could work in analogy to UV radiation. Light and in particular UV radiation provides the energy for the photochemical synthesis of biomolecules that are of interest in origin-of-life research. This can favor the formation of molecules that function in extant biology while inhibiting the formation of molecules that do not [84]. We suggest that, although the Miller–Urey experiment may not be directly transferrable to the existing prebiotic conditions of Earth, a great deal can be learned regarding the generation and the behavior of more or less spontaneously forming prebiotic broths. How to translate this understanding to the conditions at the origin will then be a different question.

## 5. Conclusions

Here, we studied the influence of temperature, ammonia concentration, spark generator and pressure on the chemical composition of a Miller–Urey-type prebiotic broth.

The choice of spark generator had the most important influence on the range and diversity of the synthesized compounds. Ammonia exhibited a catalytic role in generating non-nitrogen-containing motifs and compounds. A lower ammonia concentration could be partly accounted for by using a different spark generator. The highest temperature in our study resulted in greater chemical diversity but a lower total mass of the generated product.

## Figures and Tables

**Figure 1 life-13-02201-f001:**
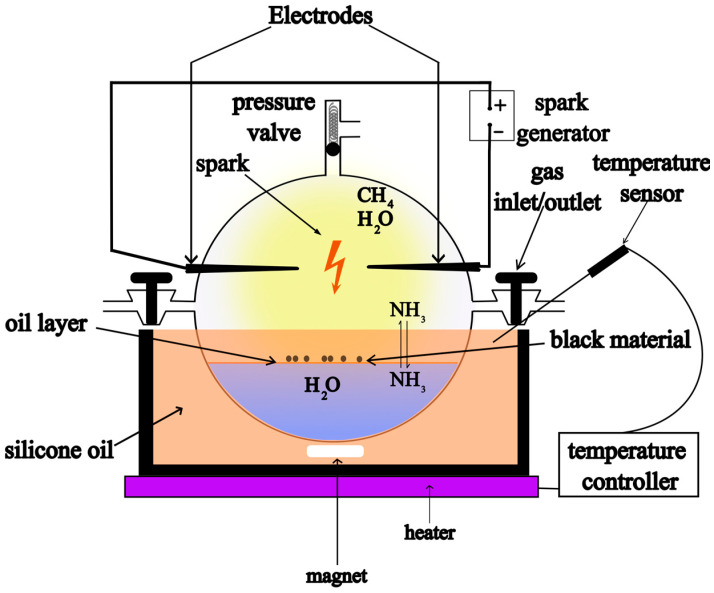
Experiment setup.

**Table 1 life-13-02201-t001:** Experimental conditions.

Experiment # ^1^	Temperature ^2^ (°C)	NH_3_ ^3^ (gr/L)	Initial Pressure ^4^ (bar)	Spark Generator	Product Weight (mg)
Sample 1	+80	5.83	1.0	FB-1	3.0
Sample 2	+100	5.83	1.0	FB-1	15.6
Sample 3	+120	5.83	1.0	FB-1	1.0
Sample 4	+100	0.11	1.0	FB-1	1.6
Sample 5	+100	11.66	1.0	FB-1	5.2
Sample 6	+100	5.83	0.7	FB-1	4.0
Sample 7	+100	5.83	1.0	FB-2	5.1
Sample 8	+100	5.83	1.0	CA-1	9.3

^1^ Indicates the number of the experiment (for reference). ^2^ Temperature refers to the silicon oil bath. ^3^ The ammonia concentration refers to the initial amount of dissolved ammonia in the liquid phase. ^4^ The initial pressures are measured at room temperature.

**Table 2 life-13-02201-t002:** Detected compounds.

Compound Name	S1	S2	S3	S4	S5	S6	S7	S8
Alkanes C_12_-C_44_	+	+	+	+	+	+	+	+
Aromatics	+	+	+	+	+	+	+	+
PAHs					+		+	+
Carbamat	+	+	+	+	+	+	+	+
Ethanimidic acid		+	+	+	+	+	+	+
Guanine		+	+	+	+	+	+	+
Aliphatic Amines	+	+	+	+	+	+	+	+
Pyridinol		+	+	+	+	+	+	
Fatty acids	+	+	+	+	+	+	+	+
m-Phenylenediamine			+	+	+	+		+
Cyanophenol			+	+	+	+	+	
Benzamide		+	+	+	+		+	
Dimethylphenol			+	+	+	+		
Phenol	+		+	+	+	+		
Urea					+	+	+	+
Oxalic acid			+		+		+	
Methoxyphenol				+	+		+	
Pyrazine-2-carboxamide				+	+			+
1,4-Benzoquinone	+	+	+		+	+	+	+
4-Pyrimidinecarboxaldehyde						+	+	+
Benzyl alcohol				+	+	+		
Fatty alcohols	+	+	+	+	+		+	
Butadyne		+	+	+				
Methylbenzamide					+		+	+
Ethyl-acridone					+		+	+
Glycolic acid	+		+	+				
Biphenyldiol					+	+	+	
Biphenylene derivative					+		+	
ethylene glycol	+	+	+		+	+	+	
PEG strands	+		+					+
Amino-O-cresol					+		+	

Note: (+) indicates that the molecule is detected in the corresponding sample.

**Table 3 life-13-02201-t003:** Fragments’ abundance.

Experiment #	Fragments *m/z* in Order of Abundance (Left to Right Decreased)									
Sample 1	73	117	75	132	313	129	57	221	43	145
Sample 2	117	73	75	132	313	129	57	43	145	71
Sample 3	117	73	75	132	313	129	57	43	145	71
Sample 4	73	221	75	117	222	263	57	43	147	132
Sample 5	73	165	75	147	180	117	221	330	175	45
Sample 6	165	73	180	75	166	149	175	117	147	43
Sample 7	117	73	341	75	132	129	145	57	43	55
Sample 8	73	165	147	75	117	180	43	45	149	166

Note: (#) indicates the number of the experiment, see Table 1 for experimental conditions.

**Table 4 life-13-02201-t004:** Fragmented ions.

Molecular Ion (*m*/*z*)	Predicted Fragment Structure	Literature and Common Compounds	Relevant Studies from a Fragment	References
43	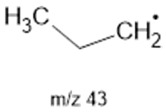	SHs ^1^	-	[35]
45	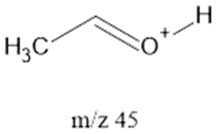	Ketons, FAcs ^2^, Fals ^3^	α β cleavage	[36,37,38]
55	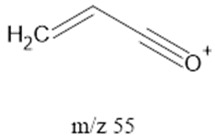	Ketones, FAcs	-	[35,39]
57	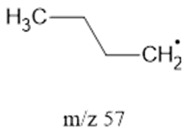	SHs	-	[35]
71	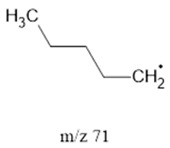	SHs	-	[35]
71	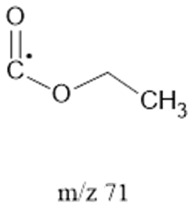	Esthers	α cleavage	[35]
71	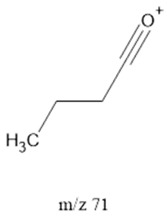	FAcs	-	[39]
73	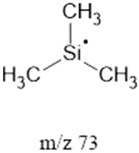	Ethers, Aldehydes	Trimethylsilyl cation	[40]
74	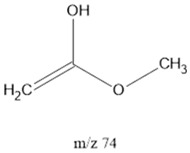	FAcs, Esthers	McLafferty rearrangement	[41]
75	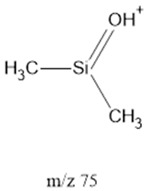	Ethers	α-fission	[42]
85	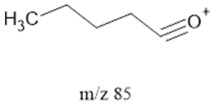	Ketones, SHs	-	[35,38,39,43,44]
117	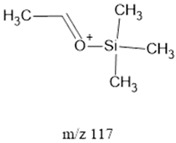	FAcs, Fals, Ethers	-	[45]
129	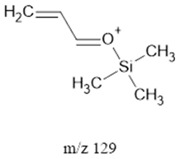	FAcs, Sterols, FAls	-	[40,46]
131	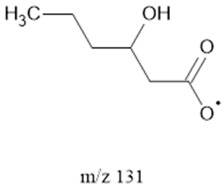	FAcs	-	[47,48]
132	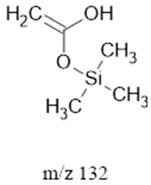	FAcs	McLafferty rearrangement	[45,49,50]
145	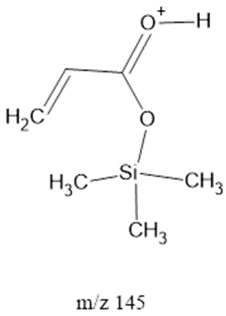	FAcs	-	[40,50]
147	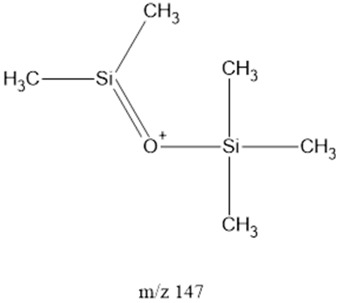	FAcs, FAls, PEG ^4^, ethers, dCAs ^5^	-	[45,51,52]
149	-	FAcs, FAls, PEG, ethers, dCAs	Hydrogenation of *m*/*z* = 147	[45,51,52,53]
165	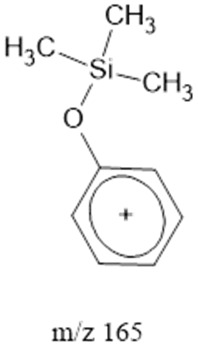	Phenols	-	[45,54,55]
166	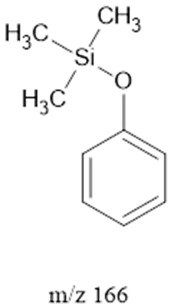	Phenols	Hydrogenated *m*/*z* 165	[45,54,55]
175	-	FAcs	Cleavage of the C3–C4 linkage	[47,56]
180	-	bisphenols	165, 180, 236, 242 for AA derivatized BPS	[45,54,55,57,58]
221	-	contamination	AMDIS	-
222	-	contamination	AMDIS	-
263	-	contamination	AMDIS	-
313	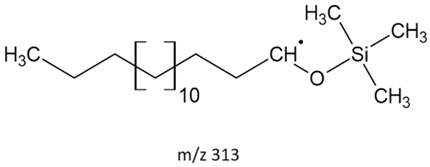	FAcs	AMDIS	[59]
330	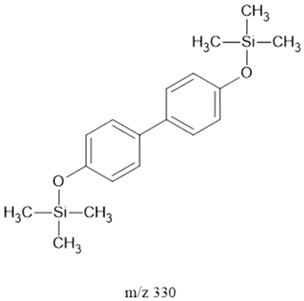	-	AMDIS	-
341	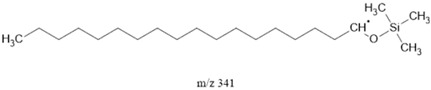	FAcs, FAls		[59]

^1^ Saturated Hydrocarbons = SHs. ^2^ Fatty Acids = FAcs. ^3^ Fatty Alcohols = Fals. ^4^ Polyethylene Glycol = PEG. ^5^ Dicarboxylic Acids = dCAs. Note: **^.^** and **^+^** represent the most likely radical and positively charged location, respectively.

## Data Availability

The data presented in this study are available on request from the corresponding author.

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
