# Peer review of "The Nature of the Spark Is a Pivotal Element in the Design of a Miller–Urey Experiment"

_life, 2023, doi:10.3390/life13112201_

Round 1

Reviewer 1 Report

Comments and Suggestions for Authors

The paper by the Ott group describes the effect of the nature of the electric discharge on the outcome of Miller-Urey type experiments.  The paper is well written and clearly shows that even small variation in the physical parameters of the spark can significantly alter the composition of the products obtained.  This is a very important point at the design of Miller-type experiments, especially knowing that the original experiment (as demonstrated in Mohammadi, E. et al., Chem. Eur. J. 2020, 26(52), 12075–12080) was performed using sparks that have entirely different physical characteristics as compared to those formed in real terrestrial lightning events.

The paper is well-written, nonetheless, in my opinion the discussion on the product compositions could be more detailed.  For example, it is hard to believe for me that no formic acid is detected in their reaction mixture, since it forms in all kinds of Miller-Urey experiments.  I would appreciate if upon revision the authors address also this point.  

Table 2 is good enough for the main text, but the authors could add some more detailed tables into the Sup info showing the chemical structures of the compounds they summarize in Table 2 as "aliphatic amines", "fatty acids", "fatty alcohols", "PAHs", and "aromatics".

Author Response

The paper by the Ott group describes the effect of the nature of the electric discharge on the outcome of Miller-Urey type experiments.  The paper is well written and clearly shows that even small variation in the physical parameters of the spark can significantly alter the composition of the products obtained.  This is a very important point at the design of Miller-type experiments, especially knowing that the original experiment (as demonstrated in Mohammadi, E. et al., Chem. Eur. J. 2020, 26(52), 12075–12080) was performed using sparks that have entirely different physical characteristics as compared to those formed in real terrestrial lightning events.

We thank the reviewer for his or her comments. We greatly appreciate the insightful feedback regarding the effects of the nature of the electric discharge in Miller-Urey-type experiments.

We included a reference to Mohammadi et al., Chem. Eur. J. 2020, 26(52) (line 48). The related paragraph now reads:

In a recent study, Mohammadi et al. proposed that the accumulation of formic acid as an intermediate of Fischer-Tropsch synthesis may well have occurred on the early Earth through multiple, independent pathways [Chem. Eur. J., vol. 26, no. 52, pp. 12075–12080, 2020 ] and [Appl. Catal. A Gen., vol. 186, no. 1–2, pp. 3–12, 1999 ] . They note that ammonium salts of formic acid have been suggested to serve as the precursor of formamide. Ferus et al. revisited the Miller-Urey experiments by using electric discharge and laser-driven plasma simulations in a reducing atmosphere to investigate nucleobase formation from formamide (Proc. Natl. Acad. Sci. U. S. A., vol. 114, no. 17, pp. 4306–4311)

The paper is well-written, nonetheless, in my opinion the discussion on the product compositions could be more detailed.  For example, it is hard to believe for me that no formic acid is detected in their reaction mixture, since it forms in all kinds of Miller-Urey experiments.  I would appreciate if upon revision the authors address also this point. 

We agree that a more detailed discussion on the product compositions will improve the manuscript. We now acknowledge the limitation of our study in terms of method development and separation capability (line 244). We now say:

Limitations of our analytical method

When assessing Gas Chromatography-Mass Spectrometry (GC-MS) in relation to other analytical tools, it's important to recognize its limitations. One significant limitation pertains to the vaporization of the analyte, which we improve through derivatization techniques. Another challenge concerns the effective separation of compounds within the GC-MS system.

We used a HP-5MS UI fused silica capillary column (Agilent Technologies) that is a conventional column widely used in research laboratories. This column is commonly employed in many labs. It is non-polar and will not provide optimal separation for polar compounds with shorter retention times. However, developing a non-targeted comprehensive method for all compounds is impossible.

Moreover, we expect several factors to contribute to a weak detection of certain compounds in our study, such as for instance formic acid or amino acids, which were detected elsewhere by others (Chem. Eur. J., vol. 26, no. 52, pp. 12075–12080, 2020).

  1. Method and Run Time: The five-day duration of our experiment may have impacted the composition by altering the ratios of components. Different run times can lead to variations in product yields and product composition.
  2. Extraction Procedure: The use of chloroform, a non-polar solvent, for extraction may exclude or reduce the recovery of highly polar compounds, including formic acid.
  3. Chemical Reactions: During the drying or extraction process, unintended reactions could have occurred, potentially altering the composition in both the water-methanol and chloroform phases [39][34].
  4. GC-MS Method: The wide range of compounds generated in our experiment posed a challenge for analysis. To ensure clarity and achieve robust results, we focused on non-polar compounds with m/z values between 50 and 550. This allowed us to report compounds with acceptable signal-to-noise ratios while ignoring peaks that did not meet our predefined standards. This occured in situations where substances could not be separated by the column because their migration properties were too similar.”

Table 2 is good enough for the main text, but the authors could add some more detailed tables into the Sup info showing the chemical structures of the compounds they summarize in Table 2 as "aliphatic amines", "fatty acids", "fatty alcohols", "PAHs", and "aromatics".

In the revised version, we introduced Table S4 in the supplementary materials, which provides a comprehensive overview of the chemical structures of various compounds, including "aliphatic amines," "fatty acids," "fatty alcohols," "PAHs," and "aromatics and others." This addition serves to elucidate the acronyms referenced throughout the main body of the paper, enhancing clarity and accessibility.

Reviewer 2 Report

Comments and Suggestions for Authors

The Miller-Urey experiment represents a miliar stone in the field of biochemistry and a lot of studies has been performed along these seven decades. In this paper the authors analyze the role played by the tipology of the electric sparks in biasing the final result. In literature other articles have addressed this kind of topic and from this point of view the present study is not so innovative. Anyway, it adds a small piece to better understand the Miller-Urey experiment and I believe that it can be useful for further researches along this direction.

However, the authors seem to completely ignore a series of studies that have reproduced the Miller-Urey experiment using purely theoretical-computational approaches. See for example:

DOI: 10.1038/NCHEM.2099; DOI: 10.1073/PNAS1402894111; DOI: 10.26434/CHEMRXIV-2022-15ct6-v3

Apart from this point there are only some minor points. The authors they need to pay more attention to the definition of acronyms: for example PAHs is defined after the acronym is defined. Moreover more technical details on the specifications of the electrodes could be useful.

In conclusion the paper after this minimal changes can be published.

Comments on the Quality of English Language

The manuscript is written in a good english.

Author Response

The Miller-Urey experiment represents a miliar stone in the field of biochemistry and a lot of studies has been performed along these seven decades. In this paper the authors analyze the role played by the tipology of the electric sparks in biasing the final result. In literature other articles have addressed this kind of topic and from this point of view the present study is not so innovative. Anyway, it adds a small piece to better understand the Miller-Urey experiment and I believe that it can be useful for further researches along this direction.

However, the authors seem to completely ignore a series of studies that have reproduced the Miller-Urey experiment using purely theoretical-computational approaches. See for example:

DOI: 10.1038/NCHEM.2099; DOI: 10.1073/PNAS1402894111; DOI: 10.26434/CHEMRXIV-2022-15ct6-v3

We thank the referee for his review, of great help in improving our manuscript. We point out that we did not find any experimental work that addresses the nature of the spark, though.

We indeed omitted the theoretical-computational work. We now gladly cite the above in the revised version. The introduction now includes the following (line 63): Theoretical computational approaches have been developed as well. A reactive machine learning system combined with a nanoreactor molecular dynamics sampler was used to simulate the behavior of C, H, N, and O elements in a wide range of real-world reactive systems. It closely matched experimental structures in carbon solid-phase nucleation and graphene ring formation studies and produces reliable predictions in cases where experimental data is not available [ DOI: 10.26434/CHEMRXIV-2022-15ct6-v3]. A different ab initio nanoreactor simulation suggested new pathways for glycine synthesis from primitive compounds proposed to exist on the early earth. However, such a nanoreactor does not aim to replicate the physicochemical conditions of any specific environment [ DOI: 10.1038/NCHEM.2099]. A theoretical study at the quantum level based on ab initio molecular dynamics showed that, rather than the activation energy of electric discharge, it was the strong electric field that favored the formation of small intermediate molecules such as formic acid and formamide in Miller-Urey type experiments [ DOI: 10.1073/PNAS1402894111].

Apart from this point there are only some minor points. The authors they need to pay more attention to the definition of acronyms: for example PAHs is defined after the acronym is defined.

We corrected for this (see line 169). We checked carefully to give acronyms only after definition of the group of substances.

As mentioned in the response to referee 1, we added the new Table S4-supplementary materials to the supplementary materials to present more structural details regarding the chemical structure of "aliphatic amines", "fatty acids", "fatty alcohols", "PAHs", and "aromatics and others". This should clarify the acronyms mentioned in the body of the paper.

Moreover more technical details on the specifications of the electrodes could be useful.

We added more technical details on the electrodes as follows (line 110):

“We utilized tungsten electrodes in our experiment. They were 30 cm in length, 6 mm in diameter, exhibited a length of their cone of 15mm and a tip with 0.5 mm radius. At the opposite end of the electrodes, there was a 20 mm deep pit with 2.5 mm internal diameter for connection of high voltage and earth cables through a ferrule embedded within the hole. The analysis using Energy Dispersive X-ray Spectroscopy (EDX) revealed the tungsten composition to be highly pure, with only trace amounts of impurities, notably Iron, Cobalt, Nickel, and Copper.”

 In conclusion the paper after this minimal changes can be published.

Reviewer 3 Report

Comments and Suggestions for Authors

Dear editors and dear authors, the manuscript given to me to review named “The nature of the spark is a pivotal element in the design of a Miller-Urey experiment” describes a modified variant of the well-known Miller-Urey experiment. The authors performed this experiment utilizing eight different conditions where the temperature and the concentration of ammonia differ with two types of electrical /spark discharge flyback-based and capacitor-based. 

The manuscript describes the experimental design, the conditions and the results in a straight and clear way which is easy to read in a well-organized manner. Materials and methods are described in an extremely detailed way (like the diameter of the electrodes) which indicates a well-planned experiment with great care.   The results are published in tables in a convenient way and described with sufficient details.

This manuscript could be published if the authors consider the following concerns.

Major concerns (mostly for the introduction and discussion of the manuscript)

The authors should describe the justification and the significance better of this experiment. Since the Miller-Urey experiment was revisited many times so far, there must be a clear statement why there is a need for new experiment? How does the circulating fluid design of the Miller-Urey experiment compare with the one pot design described by the authors?

The manuscript should reflect the geological conditions of the prebiotic earth (it is connected with prior concern). I would like to see a discussion about how more updated suggestions for prebiotic atmosphere reflect this experiment. The Authors need to address the carbon cycle model (PMID: 29610313), the oxygen level of the prebiotic atmosphere (PMID: 11537741, PMID: 32010786, PMID: 32487988, PMID: 36067299).

This experiment may contribute to the more sophisticated models for prebiotic chemistry, with those published by Sutherland and team (PMID: 33880920, PMID: 32076638). 

Where are the amino acids? From the perspective of building blocks of life, the Miller-Urey experiment and the revised modifications of this experiment described the formation of amino acids. Also, it is important to describe any precursors or actual nucleotide formation (if any). If your experiment is relevant and produced any of the building blocks (amino acids, nucleotides). I see FA was described well in the manuscript. If you have data please provide a table with the abundance level of the amino acids.

Some minor concerns:

Figure 1. Increase the size of the letters so to be able to see better.

Author Response

Dear editors and dear authors, the manuscript given to me to review named “The nature of the spark is a pivotal element in the design of a Miller-Urey experiment” describes a modified variant of the well-known Miller-Urey experiment. The authors performed this experiment utilizing eight different conditions where the temperature and the concentration of ammonia differ with two types of electrical /spark discharge flyback-based and capacitor-based. 

The manuscript describes the experimental design, the conditions and the results in a straight and clear way which is easy to read in a well-organized manner. Materials and methods are described in an extremely detailed way (like the diameter of the electrodes) which indicates a well-planned experiment with great care.   The results are published in tables in a convenient way and described with sufficient details.

We thank the referee for his review and his positive assessment.

This manuscript could be published if the authors consider the following concerns.

Major concerns (mostly for the introduction and discussion of the manuscript)

The authors should describe the justification and the significance better of this experiment. Since the Miller-Urey experiment was revisited many times so far, there must be a clear statement why there is a need for new experiment?

We introduced such a statement. (line 86)

Here we address the reactor physical parameters (i.e. the spark, temperature, ammonia concentration, and pressure) and their effect on composition. Note that these parameters are linked, i.e. the pressure and temperature change the nature of the spark. Concentrations may intervene in the pressure and the ionization of the gas mixture during sparking. To our knowledge, this set of parameters has not been investigated before, in particular the influence of the spark generator.

How does the circulating fluid design of the Miller-Urey experiment compare with the one pot design described by the authors?

We added (line 81):
In our previous findings we highlighted the reactor design as an important factor affecting the complex mixture of a Miller-Urey type experiment [33]. To maintain a high level of experimental control and facilitate a comprehensive analysis of the significance of each parameter, we streamlined the design of the traditional circulating reactor to a simplified one-pot reactor. This reduced the complexity, ensuring a high level of experimental control for data comparison. 

The manuscript should reflect the geological conditions of the prebiotic earth (it is connected with prior concern). I would like to see a discussion about how more updated suggestions for prebiotic atmosphere reflect this experiment. The Authors need to address the carbon cycle model (PMID: 29610313), the oxygen level of the prebiotic atmosphere (PMID: 11537741, PMID: 32010786, PMID: 32487988, PMID: 36067299).

This experiment may contribute to the more sophisticated models for prebiotic chemistry, with those published by Sutherland and team (PMID: 33880920, PMID: 32076638). 

We introduced the following paragraph into the discussion (line 356):

The question if the Miller-Urey experiment can be linked to the conditions on the early earth in simple ways has been ongoing since the original work. Evidently, sulfur and phosphate, two most important elements in today’s organisms, are lacking in our setup, and they may have had a strong role [56]. However, these substances could be added in future work, albeit complicating the analysis. In principle, also the gas mixture could be changed to more oxidizing mixtures that may be closer to the atmosphere of the early earth. However, such experiments were performed before to conclude that a highly reductive atmosphere was a requirement to produce a highly complex broth [57]. In this context, Bada et al. showed that the yield of amino acids is greatly increased when oxidation inhibitors such as ferrous iron are added prior to hydrolysis to counteract a more oxidizing gas mixture [58]. The conclusion that locally reduced conditions were sufficient for the generation of biomolecules challenged the belief that the earth’s atmosphere was just too oxidizing for a prebiotic broth to form spontaneously. Moreover, the oxidizing, neutral, or even slightly reducive nature of the earth’s atmosphere, as well as of its surface,4-4.5 109 years ago in a prebiotic world, as well as beyond that date remains under discussion [59][60][61].

We do not believe our spark to compare to lightning, however, it may work along similar lines as UV radiation. Light, in particular UV radiation, provides the energy for photochemical synthesis of biomolecules that are of interest in origin of life research. This could favour the formation of molecules that function in extant biology while inhibiting the formation of molecules that do not [62]. We suggest that, although the Miller-Urey experiment may not be directly transferrable to the existing prebiotic conditions of the earth, nevertheless a great deal can be learned regarding the generation and the behavior of more or less spontanously forming prebiotic broths. It will then be a different question, how to translate this understanding to the conditions at the origin.

Where are the amino acids? From the perspective of building blocks of life, the Miller-Urey experiment and the revised modifications of this experiment described the formation of amino acids. Also, it is important to describe any precursors or actual nucleotide formation (if any). If your experiment is relevant and produced any of the building blocks (amino acids, nucleotides).

We now discuss the limitation of our study in terms of applied method and separation capability due to the use of GC-MS. We introduced a paragraph to the results section entitled “Limitations of our analytical method” as follows:

“One significant limitation pertains to the vaporization of the analyte, which we improve through derivatization techniques. Another challenge concerns the effective separation of compounds within the GC-MS system.

We used a HP-5MS UI fused silica capillary column (Agilent Technologies) that is a conventional column widely used in research laboratories.. This column is commonly employed in many labs. It is non-polar and will not provide optimal separation for polar compounds with shorter retention times. However, developing a non-targeted comprehensive method for all compounds is impossible.

Moreover, we expect several factors to contribute to a weak detection of certain compounds in our study, such as for instance formic acid or amino acids, which were detected elsewhere by others [36][37][38].

  1. a. Method and Run Time: The five-day duration of our experiment may have impacted the composition by altering the ratios of components. Different run times can lead to variations in product yields and product composition.
  2. Extraction Procedure: The use of chloroform, a non-polar solvent, for extraction may exclude or reduce the recovery of highly polar compounds, including formic acid.
  3. Chemical Reactions: Chemical Reactions: During the drying or extraction process, unintended reactions could have occurred, potentially altering the composition in both the water-methanol and chloroform phases [39][34].
  4. GC-MS Method: The wide range of compounds generated in our experiment posed a challenge for analysis. To ensure clarity and achieve robust results, we focused on non-polar compounds with m/z values between 50 and 550. This allowed us to report compounds with acceptable signal-to-noise ratios while ignoring peaks that did not meet our predefined standards. This occured in situations where substances could not be separated by the column because their migration properties were too similar. “

I see FA was described well in the manuscript. If you have data please provide a table with the abundance level of the amino acids.

We now introduced Table S4 in the supplementary materials, which provides a comprehensive overview of the chemical structures of various compounds, including "aliphatic amines," "fatty acids," "fatty alcohols, amino acids, " "PAHs," and "aromatics and others." This addition serves to elucidate the acronyms referenced throughout the main body of the paper, enhancing the clarity and accessibility of our work. Regarding the detected amino acids’, we added a list out of some artificial amino acids found in the samples but to present the quantification report out of them, we needed a targeted strategy for amino acids purification and internal standards comparation for this matter. We aimed into a non-targeted strategy to evaluate the sensitivity of the reactor outcome to the altered parameters. 

Some minor concerns:

Figure 1. Increase the size of the letters so to be able to see better.

We increased the letter size of the figures.

Reviewer 4 Report

Comments and Suggestions for Authors

Conducting experiments, obtaining data and describing the results do not raise questions in the qualification of the authors. The formulation and purpose of the work raises questions. An electric discharge on early Earth implies lightning, mostly associated with volcanic eruptions. Volcanic gases then and now consist mainly of carbon dioxide and nitrogen oxides. Methane and ammonia are very rare and require special conditions for survival. But it is these gases that are absent in the experiments, which looks very artificial. Therefore, I would recommend repeating the experiments with the addition of nitrogen and carbon dioxide. Perhaps the set of products and conclusions will change significantly.

Author Response

Conducting experiments, obtaining data and describing the results do not raise questions in the qualification of the authors. The formulation and purpose of the work raises questions. An electric discharge on early Earth implies lightning, mostly associated with volcanic eruptions. Volcanic gases then and now consist mainly of carbon dioxide and nitrogen oxides. Methane and ammonia are very rare and require special conditions for survival. But it is these gases that are absent in the experiments, which looks very artificial. Therefore, I would recommend repeating the experiments with the addition of nitrogen and carbon dioxide. Perhaps the set of products and conclusions will change significantly.

We greatly appreciate the reviewer for his/her insightful feedback. We agree that the way we designed our Miller-type experiments, especially when it comes to the nature of the electric discharge as well as the mixture of gases, is a crucial aspect. We're fully aware that real lightning, often linked with volcanic eruptions, is quite different from the sparks we generated in our lab, and we appreciate the opportunity to discuss this.

Gas mixtures:

We appreciate the proposition to use different gas mixtures, and we gave this a thought, however, a reductive atmosphere is a necessary condition to produce a huge variety of molecules. The oxidative nature of the gases carbon monoxide and nitrogen oxides leads to a heavily reduced variety of the produced molecules (Orig Life Evol Biosph (2008) 38:105–115 & J Mol Evol (1983) 19:383-390). This would have to be circumvented somehow (see for instance the addition of oxidation inhibitors such as ferrous iron by Bada et al.  Orig. Life Evol. Biosph., vol. 38, pp. 105–115, 2008). We understand that, a priori, the degree of novelty of a study that concerns different gas mixtures is not clear. At the same time our gas mixture is close to the original Miller-Urey experiment. The original experiment has been analyzed at length with different techniques by others. Placing our work in vicinity makes it comparable up to a point.  We also point towards a great deal of uncertainty about what the early Earth's atmosphere was like. Even estimating the abundance of carbon dioxide has evaluated to be quite tricky, as recent research (Sci. Adv., vol. 6, no. 4, p. eaay4644, 2020) has shown, and the lack of preserved data from the early Earth makes things even more challenging for our case. A practical problem is that introducing different gas mixtures as another parameter is multiplicative with the number of experimental points. Since each run of each experiment takes a week to perform, we would quickly reach time scales of months (if not years) for an exhaustive study.

Spark:

The spark is a convenient means to generate a highly complex prebiotic broth at lab scale. This does not mean that the experiment is directly comparable to the early earth. The spark can be replaced by other means, light or a strong temperature difference (Chem.Eur.J.2016,22,3572–3586) for instance. However, the conclusions from our work remain the same, it is the timing of the effective temperature levels of the spark that will have a strong impact on the composition of the broth. We believe to be the first to point this out with clear underlying data.

In order to make these points clear, we added the following to the discussion (Line 353):

The question if the Miller-Urey experiment can be linked to the conditions on the early earth in simple ways has been ongoing since the original work. Evidently, sulfur and phosphate, two most important elements in today’s organisms, are lacking in our setup, and they may have had a strong role [56]. However, these substances could be added in future work, albeit complicating the analysis. In principle, also the gas mixture could be changed to more oxidizing mixtures that may be closer to the atmosphere of the early earth. However, such experiments were performed before to conclude that a highly reductive atmosphere was a requirement to produce a highly complex broth [57]. In this context, Bada et al. showed that the yield of amino acids is greatly increased when oxidation inhibitors such as ferrous iron are added prior to hydrolysis to counteract a more oxidizing gas mixture [58]. The conclusion that locally reduced conditions were sufficient for the generation of biomolecules challenged the belief that the earth’s atmosphere was just too oxidizing for a prebiotic broth to form spontaneously. Moreover, the oxidizing, neutral, or even slightly reducive nature of the earth’s atmosphere, as well as of its surface,4-4.5 109 years ago in a prebiotic world, as well as beyond that date remains under discussion [59][60][61].

We concur with the findings of Mohammadi et al [23]. Nevertheless, our sparks cannot be compared to lightning, however, they could work in analogy to UV radiation. Light, in particular UV radiation, provides the energy for photochemical synthesis of biomolecules that are of interest in origin of life research. This can favour the formation of molecules that function in extant biology while inhibiting the formation of molecules that do not [62]. We suggest that, although the Miller-Urey experiment may not be directly transferrable to the existing prebiotic conditions of the earth, nevertheless a great deal can be learned regarding the generation and the behavior of more or less spontanously forming prebiotic broths. It will then be a different question, how to translate this understanding to the conditions at the origin.

Round 2

Reviewer 3 Report

Comments and Suggestions for Authors

Thank you for your revised manuscript. Now your experiment seems better justified and explained.

I think your experiment will contribute to the field of prebiotic synthesis, so it will be nice to see your manuscript published.

Reviewer 4 Report

Comments and Suggestions for Authors

Of course, the origin of life on Earth took place in very difficult and unknown conditions. Authors need to try a lot of options and not focus on just one. As development of the Miller & Urey experiment this work certainly deserves attention and publication.